# Microautophagy in Plants: Consideration of Its Molecular Mechanism

**DOI:** 10.3390/cells9040887

**Published:** 2020-04-04

**Authors:** Katarzyna Sieńko, Andisheh Poormassalehgoo, Kenji Yamada, Shino Goto-Yamada

**Affiliations:** Malopolska Centre of Biotechnology, Jagiellonian University, 30-387 Krakow, Poland; katarzyna.sienko@uj.edu.pl (K.S.); a.masalehgou@gmail.com (A.P.); kenji.yamada@uj.edu.pl (K.Y.)

**Keywords:** microautophagy, autophagy, plant, autophagy-related genes, starvation, vacuole, lysosome, stress response, degradation, organelle

## Abstract

Microautophagy is a type of autophagy. It is characterized by direct enclosing with the vacuolar/lysosomal membrane, which completes the isolation and uptake of cell components in the vacuole. Several publications present evidence that plants exhibit microautophagy. Plant microautophagy is involved in anthocyanin accumulation in the vacuole, eliminating damaged chloroplasts and degrading cellular components during starvation. However, information on the molecular mechanism of microautophagy is less available than that on the general macroautophagy, because the research focusing on microautophagy has not been widely reported. In yeast and animals, it is suggested that microautophagy can be classified into several types depending on morphology and the requirements of autophagy-related (*ATG*) genes. This review summarizes the studies on plant microautophagy and discusses possible techniques for a future study in this field while taking into account the information on microautophagy obtained from yeast and animals.

## 1. Introduction

Autophagy, so-called “self-eating” of the cell, is the main pathway for intracellular degradation and the recycling system. Recent studies have shown that autophagy contributes to various plant developmental processes and responses to stress, such as nutrient starvation and infections, and it is clear that autophagy is essential for maintaining cellular homeostasis [1,2,3,4]. So far, three main types of autophagy have been differentiated depending on the mechanism through which cargo is directed to the vacuole (in yeast and plants) or the lysosome (in animals) for degradation: macroautophagy, microautophagy, and chaperone-mediated autophagy [5,6]. Macroautophagy is the most well-studied type of autophagy, and more than 40 autophagy-related (ATG) proteins have been identified as contributing to macroautophagy in various eukaryotes [4,7,8]. A double-membrane vesicle (autophagosome) is formed to enclose a portion of the cytoplasm with cellular compartments or selected cargo, such as organelles. Then the vesicle fuses to the vacuolar/lysosomal membrane with its outer membrane, which is followed by the release of the cargo surrounded by the inner membrane into the lumen for digestion and recycling. Microautophagy, on the other hand, is the direct uptake of cargoes into the vacuole/lysosome by the invagination or protrusion of the vacuolar/lysosomal membrane [9]. Although the term “microautophagy” was first used in 1966 by de Duve and Wattiaux to describe the process of cytoplasm sequestration and subsequent engulfment by lysosomes in mammalian cells, there is to date less information in the literature on microautophagy compared with that on macroautophagy [9,10,11].

Oku and Sakai (2018) proposed categorizing microautophagy into three types depending on the morphology of the vacuolar/lysosomal membrane deformation: (i) with the lysosomal/vacuolar membrane protrusion, (ii) invagination, and lately discovered (iii) invagination of late endosomes [12]. Additionally, selective or non-selective microautophagy was differentiated based on the selectivity of the engulfment of cargoes during these processes; the cargo may consist of non-selective soluble components in the cytosol [13,14], however, particular organelles and cellular components can also be sequestered in selective microautophagy, such as micropexophagy (peroxisome) [15,16,17], piecemeal microautophagy of the nucleus (PMN, part of the nucleus) [18,19], micromitophagy (mitochondrion) [20], and microlipophagy (lipid droplet) [21,22,23].

In this review, we will focus on the recent findings in the field of microautophagy that emphasize microautophagy in plants and will provide a description of the morphological and molecular basis of this process. Furthermore, we would like to summarize the factors that are reported to be involved in microautophagy regulation in yeast and animals and find their homologs in plants as potential markers for future analysis.

## 2. Microautophagy in Yeast and Animals

### 2.1. Microautophagy in Yeast—a Model for Plant Studies

Since the discovery of *ATG* genes in *Saccharomyces cerevisiae* (baker’s yeast), the number of studies on autophagy involving yeast as a model species has grown rapidly. The majority of papers on microautophagy present results that are based on yeast species. The information gained from these studies provides an excellent baseline for further research to be performed on other organisms. As a consequence of these existing studies, we can now say more about the molecular mechanisms that underlie microautophagy and its biological importance in organisms.

*Pichia pastoris* is a methylotrophic yeast and was used in the study of peroxisome degradation via macro- and microautophagy, because peroxisome degradation can be easily controlled by media change. Tuttle and Dunn (1995) proposed the existence of two independent pathways of peroxisome degradation in *P*. *pastoris* by describing the morphological changes of peroxisomal sequestration using TEM (transmission electron microscope) observation [24]. The first pathway is the sequestration of individual peroxisomes by autophagosomes like in macroautophagy, which is induced by transferring cells from a methanol- to an ethanol-containing media. The second pathway is referred to as the engulfment of a cluster of peroxisomes with finger-like protrusion of the vacuolar membrane in a process like microautophagy, which is induced by changing the carbon source from methanol to glucose. The direct incorporation of peroxisomes into the vacuole by microautophagic invagination is defined as micropexophagy, and a number of mutants defective in micropexophagy have been isolated and analyzed by several groups (reviewed in [25]). The enclosure of a cluster of peroxisomes requires an additional membrane called the MIPA (micropexophagy-specific membrane apparatus), which contains Atg proteins and facilitates the terminal enclosure and fusion steps. Many genes are shared between macro- and micropexophagy, although several genes seem to be specific for micropexophagy [26].

Degradation of a portion of nucleolus proteins via microautophagy, which is referred to as piecemeal microautophagy of the nucleus (PMN), is reported in *S. cerevisiae* [18,19]. The nucleus-vacuole junction (NVJ) is formed by the direct interaction between the Vac8 protein on the vacuolar membrane and the Nvj1 protein in the outer nuclear envelope in response to nutrient starvation [27]. A portion of the nucleus forms a bleb at the NVJ and invaginates towards the vacuolar lumen. This vesicle, which contains nucleolar proteins, is released into the vacuolar lumen to be degraded [18,19]. PMN is characteristic of *S. cerevisiae* and has not been defined in other organisms. PMN requires the core and some specific *ATG* genes and components of the phosphoinositide 3-kinase (PI3K) complex, which are considered to be essential for terminal vacuole enclosure and the fusion stage of PMN. Although completion of the PMN process requires a part of the components essential for homotypic vacuole fusion (such as Sec17, Sec18), not all components involved in the vacuole fusion are required. Therefore, there is still a possibility that these proteins may not directly contribute to PMN, but function in vacuolar biogenesis [19].

Treatment of cells with ER (endoplasmic reticulum) stress inducers, such as DTT (dithiothreitol), tunicamycin, and CPA (cyclopiazonic acid), induces ER stress that activates ER-phagy. Schuck et al. (2009, 2014) pointed out that ER-phagy may occur via either of the macro- and microautophagic pathways in *S. cerevisiae* [11,28]. In contrast to macro-ER-phagy, the stacks of cisternal ER (ER whorls) are formed and engulfed by the vacuolar membrane in the micro-ER-phagy process. Interestingly, micro-ER-phagy does not require the core autophagy machinery, as the Nem1–Spo7 phosphatase complex and the ESCRT (endosomal sorting complex required for transport) machinery mediate micro-ER-phagy [11,29]. ESCRT proteins may participate in the scission of the lysosomal membrane to complete the microautophagic uptake of the ER into the vacuole.

Microautophagy also plays an important role in lipid droplet (LD) degradation in the yeast *S. cerevisiae*. LDs are formed in cells at the stationary phase, after the diauxic shift, during nitrogen starvation, or by inhibition of phosphatidylcholine (PC) biosynthesis at the ER, and LDs are transported to the vacuole via the microautophagy process [21,22,23,30]. It seems that lipid droplet degradation is induced to obtain new energy by decomposing unnecessary membrane components, such as during starvation, and to avoid abnormal lipid metabolism stress.

### 2.2. Regulatory Factors

Early analysis using mutants in yeasts identified the involvement of the *ATG* genes that are common in macroautophagy. Although the *ATG* genes seem to be one of the key factors in microautophagy, the recent studies have reported the involvement of factors other than ATGs, and the existence of an ATG-independent pathway. This chapter summarizes the microautophagy-related factors identified in yeast and animals.

*ATG*s are identified as the genes involved in the macroautophagy process in yeasts, and most of them are conserved in higher eukaryotes. The roles of the core ATG proteins are classified in the following steps; (i) the Atg1/ULK complex (Atg1, Atg11, Atg13, Atg17, Atg29, and Atg31), (ii) the PtdIns 3-kinase (PI3K) complex (Vps34, Vps15, Vps30/Atg6, and Atg14), (iii) Atg9 and its cycling system (Atg2, Atg9, and Atg18), and (iv) two ubiquitin-like (Ubl) conjugation systems for Atg12 (Atg5, Atg7, Atg10, Atg12, and Atg16) and Atg8 (Atg3, Atg4, Atg7, and Atg8) (reviewed in [31]). Micropexophagy requires a series of ATG proteins during the initiation, target recognition, and sequestration of peroxisomes by the vacuolar membrane and the MIPA. The terminal vacuole enclosure and the fusion stage of the PMN requires core *ATG* genes (Table 1). It is also reported that the ubiquitin-like conjugation systems participate in starvation-induced non-selective and glucose-induced selective microautophagy [32]. The importance of ATG has also been demonstrated in endosomal microautophagy (eMI) in animals, in whom the cytosolic cargo proteins are captured into intraluminal vesicles formed by invagination of the membrane of the late endosome instead of by the vacuole/lysosomes. It has been shown that Atg1 and Atg13, but not Atg5, Atg7, and Atg12, are required for eMI in *Drosophila melanogaster* [33].

The involvement of ATGs in LD degradation is somewhat complicated. As described above, in microlipophagy during the stationary phase and in starved yeast cells, membrane rafts are formed on the vacuolar membrane where they are enriched with sterols and sphingolipids. The freeze-fracture electron microscopy technique revealed that LDs, which are formed during the stationary phase or nitrogen starvation, enter into the vacuole via this domain [23]. Yeast homologs of mammalian Niemann–Pick type C proteins (NPC), Ncr1 and Npc2, are required for the formation of membrane rafts to transport sterols from the vacuolar lumen to the vacuolar membrane. The core ATG proteins are required in this lipophagy, although the possibility of indirect involvement has been suggested; lipid components of rafts are transported to the vacuoles via macroautophagy that is dependent on ATG [23,34]. On the other hand, ATG proteins are dispensable in the microautophagic degradation of LDs induced by the inhibition of PC biosynthesis or the diauxic shift, but the ESCRT complexes are essential for these processes [21,30]. Therefore, microautophagy with the same decomposition target appears to be the same, but the factors involved in each microautophagy may be different; it is difficult to say which factors are common to all types of microautophagy, and this requires careful study.

The ESCRT machinery mediates membrane remodeling processes such as membrane curvature and scission, which is required for endosomal intraluminal vesicle formation and to form multivesicular bodies (MVB), cell division, and viral budding. The ESCRT machinery consists of cytosolic protein complexes, ESCRT-0 to -III and ESCRT-associated factors (such as the Vps4-Vta1 hetero-oligomer). The ESCRT and its components are reported to be involved in several types of microautophagy (Table 1). eMI proceeds under the action of the ESCRT machinery together with the selective receptor Hsc70 or Nbr1 proteins [14,58,60]. ESCRT-I to -III are conserved in plants, and the functional analogue of ESCRT-0 and plant-specific ESCRT components are also reported [48,49,51].

TOR (target of rapamycin) is a protein kinase and known to monitor and integrate cellular energy, nutrient, and stress-related signaling. TOR activation leads to negative regulation of general macroautophagy in yeast, animals, and plants [63,64,65], while TOR functions as a positive regulator for microautophagy together with the vacuolar membrane localized EGO (exit from rapamycin-induced growth arrest) complex in *S. cerevisiae* via counterbalancing the macroautophagy-mediated membrane influx toward the vacuolar membrane [66]. However, the genes making up the EGO complex (*Gtr1*, *Gtr2*, *Meh1* and *Slm4*) in yeast are not conserved in plants, and the possibility that TOR is involved in their microautophagy regulation is still not clear. VTC (vacuole transporter chaperone) complex has been identified as a sorting machinery located on vacuolar membranes to translocate membrane proteins. Recent reports revealed that it is required for microautophagy in yeast cells; during nutrient starvation, the components of the VTC complex are concentrated on the vacuolar membrane, and deletion of these genes results in a significant reduction of the microautophagic activity [46]. However, similarly to EGO, the genes encoding the VTC components are not conserved in plants.

## 3. Microautophagy in Plants

### 3.1. Microautophagy Reported in Plants

To date, over 30 *ATG* genes have been identified in plants, and their fundamental functions are conserved, and it has been demonstrated in many papers that ATG proteins play crucial roles in the macroautophagy process [2,3,67]. However, how they are involved in microautophagy is still unclear.

Anthocyanins are water-soluble flavonoid pigments synthesized in the cytoplasm and stored in the vacuole. Chanoca et al. (2015) reported that the microautophagy mechanism was observed in the process of transport of cytoplasmic anthocyanin aggregates from the cytosol to the vacuole in *Arabidopsis thaliana* and *Eustoma grandiorum* [68]. Anthocyanin aggregates in close contact with the vacuolar membrane are directly incorporated into the vacuole. Transmission electron micrographs revealed that anthocyanin aggregates were engulfed by the protrusion of the tonoplast and the microautophagic structures formed during this process (Figure 1). The authors showed that the ATG5 protein, PI3K complex, and the SNARE component involved in vacuolar fusion are not included in the translocation of anthocyanin aggregates to the vacuole.

On the other hand, ATG proteins have been shown to be essential in chloroplast elimination via microautophagy in plants. Nakamura et al. (2018) reported that chloroplasts in *Arabidopsis* mesophyll cells were damaged and swollen by exposure to high visible light. The photodamaged chloroplasts are invaginated by the tonoplast and translocate into the vacuole via a microautophagy process, namely, chlorophagy (Figure 1) [69]. GFP-ATG8-labelled structures were localized at the swollen chloroplasts after the damage, but they never covered the entire chloroplast. The damaged chloroplast is enclosed by the tonoplast and is associated with the ATG8-containing membrane structure, which implicates the MIPA structure formed during micropexophagy to sequester peroxisome aggregates in yeast cells [25,70]. The *atg5* and *atg7* mutants were not able to complete the enclosure of the chloroplast, suggesting that the core ATGs are required for the completion of chlorophagy. It was assumed that chlorophagy might involve a specific mechanism and membrane dynamics to incorporate such a large organelle as a chloroplast.

Microautophagy is also observed in tobacco BY-2 cells and *Arabidopsis* roots during sucrose starvation (Figure 1) [71]. Fluorescence dye FM4-64 was used for visualization of the dynamics of the tonoplast to detect the microautophagy pathway under a fluorescent microscope. Tubular structures on the tonoplast, presumably microautophagy-related structures, were observed in the starved BY-2 cells, while cytosolic acid granules, which are induced by starvation, were captured by the tubular structures on the tonoplast. Papain family protease inhibitors E-64c and E-64d inhibit degradation of cellular materials in plants. Importantly, these inhibitors induce accumulation of acidic vesicles, namely, E-64d vesicles, along the tonoplast in barley and *Arabidopsis* roots and BY-2 cells in sucrose starvation [71,72,73]. When only the tonoplast was stained with FM4-64 in BY-2 and *Arabidopsis* root cells, the formed E-64d vesicles were stained with FM4-64 [71]. This result indicates that vesicles originated from the tonoplast by the microautophagic pathway. Moreover, the area of the tonoplast decreased in the cells treated with E-64d, suggesting that the tonoplast is absorbed into the E-64d vesicles. Furthermore, the amount of the tonoplast decreased in the cells treated with E-64d, suggesting that the tonoplast is taken up into the E-64d vesicles. This observation demonstrates that the microautophagy process may contribute to the vacuolar formation [71]. The number of E-64d vesicles is reduced in the roots of *Arabidopsis atg2*, *atg5*, and *atg7* mutants. The results indicate that ATG2, ATG5, and ATG7 are involved in this microautophagy. However, the contribution of these ATGs to microautophagy may be indirect, because the accumulation of E64-d vesicles is not entirely stopped in *atg* mutants. As mentioned above, macroautophagy contributes to microautophagy by providing the lipid rafts required for the initiation of microautophagy on the vacuolar membrane in yeast [23]. In fact, plants exhibit several types of microautophagy, however, more detailed analyses at a molecular level are required in the future studies.

### 3.2. Methods for Microautophagy Observation

In macroautophagy, the outer membrane of autophagosomes fuses with the tonoplast, and then the autophagic bodies with the inner membrane are delivered into the vacuole. In contrast to this, in microautophagy, autophagic bodies are directly surrounded by the membrane from the tonoplast. Therefore, the membrane’s origin is different between autophagic bodies from macroautophagy and microautophagy. If the membrane of autophagic bodies contains the vacuolar membrane from the tonoplast, it means that the autophagic bodies are derived from microautophagy (Figure 1).

Microscopic observation is one of the most useful methods to monitor structures and progression of the microautophagy process at various stages. For instance, TEM is effective in detailed morphological analyses. TEM was used for the observation of microautophagic structures in the dying nucellar cells of a flower of *Pinus densiflora* [74], and it was observed that the vacuolar membrane invaginates cellular materials for their degradation, which generates the nutrient for pollen tube growth. TEM also showed the role of microautophagy in vacuole development in a study of mesophyll cells in tepals of a *Dendrobium* orchid [75]. The vacuolar volume increases in the mesophyll cells before flower opening, and TEM observation suggested involvement of autophagy in this event. The authors detected evidence of macroautophagy together with a few signs of microautophagy with the tonoplast protruding outward at the early stage of vacuole formation. Similar observations were made in the study of dark-induced leaf senescence in barley [76]. Although another approach (such as using membrane labelling) is required for TEM analyses, detailed observation of the vacuolar form can provide one of the strong pieces of evidence of microautophagy.

A laser scanning microscope (LSM) enables the capture of living samples to study cellular dynamics and is helpful in monitoring microautophagy-associated processes, such as tonoplast remodeling, and it can suggest the origin of microautophagic vesicles. Green fluorescent proteins (GFP) tagged with a protein on the tonoplast can be one of the fluorescent markers used to visualize the tonoplast movements during microautophagy in plants. In a study on chlorophagy in *Arabidopsis*, the GFP-tagged δ-tonoplast intrinsic protein (GFP-δTIP) was used to show the evidence of microautophagy [69]. Membrane dye FM4-64 has been successfully applied to detect tonoplast dynamics during microautophagy. FM4-64 has been primarily used to monitor endocytosis in animal, yeast, and plant cells [77,78,79]. First, the dye binds to the plasma membrane and enters the cell via endocytosis. After incorporation, it is associated with the endosomal compartments and then eventually stains the tonoplast. Thus, FM4-64 requires a long incubation time to visualize the tonoplast with the dye [80]. In tobacco BY-2 cells, around 20 h of incubation are required to visualize the tonoplast with FM4-64. Because FM4-64 is a red fluorescent dye, it can be used with other types of fluorescent dyes and markers, such as GFP and quinacrine. Quinacrine is used for the staining of acidic organelles such as vacuoles. In the methylotrophic yeast (*P. pastoris*), microautophagy can be monitored with the GFP fused to type I peroxisomal targeting signal (GFP-PTS1), which is localized in peroxisomes, and FM4-64.

In plants, protease inhibitor E-64 and its analogues (e.g., E-64d, E-64c) stop degradation of autophagic bodies in the vacuole [71,73,79,81,82]. Additionally, the V-ATPase (vacuolar-type H^+^-ATPase) inhibitor, concanamycin A (ConA), reduces vacuolar hydrolase activity by stopping acidification of the vacuole, and then causes accumulation of microautophagic bodies [71]. These inhibitor treatments will be useful in distinguishing between macroautophagy and microautophagy, because they make it easy to observe autophagic bodies.

GFP-ATG8 is frequently used to monitor macroautophagy processes in yeast, animal, and plant cells, because ATG8 is crucial for the autophagosomal membrane formation. However, it is difficult to say that GFP-ATG8 is an ideal marker for microautophagy, because ATG8 is anchored on the autophagosomal membrane, but not on the tonoplast. Indeed, GFP-ATG8a co-localizes to the swollen chloroplasts that will be removed by microautophagy in *Arabidopsis*, and it seems to be true that ATG8 is involved in some of microautophagy. However, ATG8-containing membrane structures never merge with the tonoplast that is engulfing the chloroplast [69]. Besides, it appears that GFP-ATG8 is not always co-localized with the autophagic bodies derived from microautophagy in the vacuoles of *Arabidopsis* roots [71]. These observations suggest that ATG8 facilitates microautophagy, but it is still unclear whether it is indispensable for the membrane engulfment step of microautophagy. Although ATG8 is successfully utilized in various aspects of autophagy, it is important to combine different methods to reach a proper conclusion.

## 4. Conclusions

There is much evidence of microautophagy that has been shown in animals and, primarily, yeasts, but there is less evidence regarding plants [68]. However, identification of microautophagy and related structures is possible by TEM and fluorescent microscope analysis of the tonoplast markers in plants. There are several ways to visualize tonoplasts under fluorescent microscopy, e.g., FM4-64, which enables us to perform a detailed molecular analysis of microautophagy. However, we still do not know the crucial function of ATG proteins in microautophagy in plants, although it is quite well described in macroautophagy [5]. More detailed analysis of plant microautophagy may reveal unexpected aspects of its contribution to the development of plants and their response to stress. Furthermore, future studies on plant microautophagy have the potential to discover improved methods for agricultural crop plant growth.

## Figures and Tables

**Figure 1 cells-09-00887-f001:**
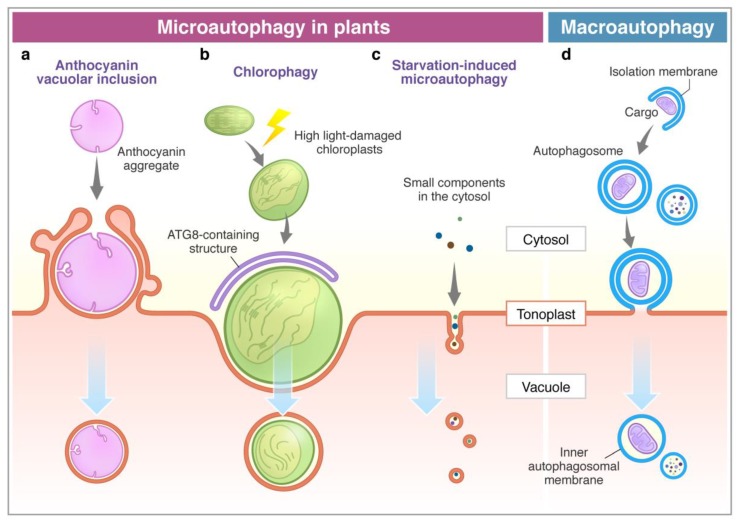
Microautophagy processes in plants. Comparison between microautophagy (**a**–**c**) and macroautophagy (**d**). (**a**) Cytosolic anthocyanin aggregate is surrounded by the protrusion of the vacuolar membrane and sequestered into the vacuole [68]. (**b**) High light-damaged swollen chloroplast invaginates toward the vacuole with the association of the ATG8-containing structure. This process requires ATG genes (ATG5 and ATG7) [69]. (**c**) Sucrose starvation induces tubule formation on the tonoplast, and then vesicles are generated. Small components, such as cytosolic proteins and cytosolic acid granules, can be transported into the vacuole via this process. This process is suppressed by lack of ATG2, ATG5, and ATG7 [71]. The membrane of autophagic bodies generated via microautophagy consists of the tonoplast (**a**–**c**). (**d**) The macroautophagy process is initiated with the enclosure of cargoes by the isolation membrane in the cytosol to form an autophagosome. After the fusion of the outer membrane of the autophagosome with the tonoplast, it generates an inner membrane-surrounded autophagic body.

**Table 1 cells-09-00887-t001:** Microautophagy-related factors and their homologs in *Arabidopsis*.

Factors Reported to be Involved in Microautophagy *		Roles in Microautophagy	Reference	Homologs in *Arabidopsis*
**ATG**	**Micropexophagy (yeast):** Atg1–5, Atg7–9, Atg11, Atg16, Atg17, Atg18, Atg21, Atg23, Atg24, Atg26, Atg28, Atg30, Atg35	Recognition of peroxisomes. Formation of the MIPA. Isolation of peroxisomes and transportation of the cargo into the vacuole.	[15,16,17,35,36,37]	ATG1–16, ATG18, ATG101 [38,39,40]
	**PMN (yeast):** Atg1–18, Atg21, Atg24, Atg29, Atg31	Enclosure of the terminal vacuole and fusion.	[18,19]	
	**Microlipophagy (yeast):** Atg1–10, Atg12, Atg14–18	Involved in the internalization of lipid droplets. Degradation of lipids by vacuolar lipase Atg15. Involved in microdomain formation during microautophagy in the stationary phase and nitrogen starvation. Involved in proper NPC distribution on the vacuolar membrane.	[22,23,41]	
**PI3K complex ****	Vps15, Vps34	Generate PI3P on the membrane to trigger microautophagy in micropexophagy, microlipophagy.	[16,22,42]	VPS15, VPS34 [43,44]
**Vacuolar membrane protein**	Vac8	Involved in vacuolar membrane fusion. Contributing to micropexophagy, PMN. Forms the nucleus-vacuole junction by binding with nuclear envelope protein Nvj1 during PMN.	[18,19]	N.I.
**Niemann–Pick type C proteins (NPC)**	Ncr1, Npc2	Form membrane rafts on the vacuolar membrane by transporting sterol during microlipophagy both in the stationary phase and in acute nitrogen starvation in yeast.	[23]	NPC1 [45]
**VTC complex**	Vtc1, Vtc2, Vtc3, Vtc4	Involved in the tubule formation on the vacuolar membrane. Recruited on the vacuole in nitrogen starvation. Directly binds to calmodulin in microautophagy in yeast.	[46]	N.I.
**ESCRT complex and related proteins**	**ESCRT-0:** Hse1, Bro1, Vps27 **ESCRT-I:** Vps23, Vps28, Vps3, Mvb12 **ESCRT-II:** Vps22, Vps25, Vps36 **ESCRT-III:** Vps2, Vps20, Vps24, Snf7, etc. **VPS4-VTA1:** Vps4	Expected to contribute to membrane bending, remodeling, and scission during micro-ER-phagy in yeast and endosomal microautophagy in animals. Full activity of Vps27 (binding to ubiquitin, PI3P, and ESCRT-I) is required for diauxic shift-induced lipophagy (yeast). Formation of lipid domains on the vacuolar membrane in response to nutrient deprivation (yeast).	[14,29,30,47]	ESCRT-I to -III, VPS4–VTA1, functional analogue of ESCRT-0, plant-specific ESCRT [48,49,50,51]
**Nem1–Spo7 complex**	Nem1, Spo7, Pah1	Expected to contribute to membrane remodeling with ESCRTs during micro-ER-phagy in yeast.	[29]	PAH1,2 [52]
**Clathrin**	Chc1	Interacts with Vps27 (ESCRT-0). Required for microlipophagy in the diauxic shift (yeast).	[30]	CHC1, CHC2 [53]
**Vacuole fusion**	**SNARE:** Vam3, Vam7 **SNAP:** Sec17 **NSF:** Sec18 **HOPS:** Vps18, Vps33, Vps39, Vps41	Can be involved in membrane fusion processes. Required for PMN and micro-ER-phagy.	[19,30]	SNARE and HOPS complexes, including α-SNAP/SEC17, NSF/SEC18, SYP22/VAM3, VPS18, 33, 39, 41 [54,55,56,57]
**Selective receptor of autophagy**	Nbr1	Selective receptor of cytosolic ubiquitinated cargo in endosomal microautophagy.	[58]	NBR1 [59]
	Hsc70	Selective receptor of cytosolic proteins in endosomal microautophagy.	[14,60]	Cytosolic/nuclear HSC70-1 to -5 [61,62]

* Indicating only the factors reported to be involved in microautophagy; ** ATG6/VPS30 and ATG14 are indicated in the ATG group; N.I., not identified. List of abbreviations: ESCRT (endosomal sorting complex required for transport), HOPS (homotypic fusion and vacuole protein sorting) complex, MIPA (micropexophagic membrane apparatus), NSF (N-ethylmaleimide-sensitive factor), PI3K (phosphoinositide 3-kinase), PI3P (phosphatidylinositol 3-phosphate), PMN (piecemeal microautophagy of the nucleus), SNAP (synaptosomal-associated protein), SNARE (soluble NSF attachment protein receptor), VTC (vacuolar transporter chaperone).

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
