# Peer review of "Microautophagy in Plants: Consideration of Its Molecular Mechanism"

_cells, 2020, doi:10.3390/cells9040887_

Round 1

Reviewer 1 Report

The manuscript by Sienko et al. nicely summarizes autophagy in eukaryotes in general and plant microautophagy with taking into account information from other eukaryotes. I believe that the manuscript is very informative to readers who would like to capture what has been known in this relatively less-explored field of plant microautophagy. I only have a couple of minor suggestions as described below.

The authors often simply use the term “starvation” but it would be better to specify the kind of starvation whenever possible.

What is “Small components in the cytosol” in Figure 1c? I recommend that the authors specify what small components refer to in the main text or in the figure legend.

L.68, TEM needs to be spelled out. Also ER in L.92.

Author Response

Answers to Reviewer 1

[The indicated line numbers are those in the "final" view without markups.]

Minor points:

(1) [Reviewer’s comment]

The authors often simply use the term “starvation”, but it would be better to specify the kind of starvation whenever possible.

[Our response]

We added the information of the type of starvation where possible, as listed below:

Line 104: nitrogen starvation

Line 134: nitrogen starvation

Table 1: nitrogen starvation (in "Microlipophagy" and "VTC complex")

Line 199: sucrose starvation

Line 207: sucrose starvation

(2) [Reviewer’s comment]

What is “Small components in the cytosol” in Figure 1c? I recommend that the authors specify what small components refer to in the main text or in the figure legend.

[Our response]

We specified what small components are by adding this information in the legend of Figure 1 (Line 200 - 201).

(3) [Reviewer’s comment]

L.68, TEM needs to be spelled out. Also ER in L.92.

[Our response]

We added this information in Line 69: TEM (transmission electron microscope) and Line 94: ER (endoplasmic reticulum).

Reviewer 2 Report

This review is timely, authoritative and interesting. It describes some roles of microautophagy in plants (and other systems). I find the description of other systems important. Furthermore, it describes some relevant methodological aspects. Although the content is important, the language suffers from poor English. 

Major points:

  1. As aforementioned, the language is difficult to read due to poor English use. I suggest serious language improvement.
  2. Authors do not present a critical evaluation of published results and in places text (described below in minor comments) is incomprehensible.
  3. In most cases, the description of the published data is swallowed.
  4. The text is very wordy and can be significantly reduced to improve readability.
  5. The message of which is the major and undisputed difference between macro- and microautophagy is not conveyed by the text. However, the image is well-prepared.
  6. The layout of the text could be better and the text better organized in distinct sessions with clearer messages (including critical evaluations in places-see also comment 2). 

Minor points:

  1. Line 15: "In yeast and animals,"
  2. Line 26: "developmental"
  3. Line 51: "In this review, "
  4. Line 62: "consequence of these existing studies, "
  5. Line 73: "Peroxisome degradation in the microautophagic manner is called micropexophagy", English here should be improved significantly.
  6. Table 1: "Envelope", "Calmodulin", "shift-induced"
  7. Line 177: "component involved in"
  8. Line 182: "photodamaged chloroplasts are invaginated by the tonoplast and 
  9. Line 203: Reference missing
  10. Line 209: "Arabidopsis roots and sucrose"
  11. Line 213: "with E-64d, suggesting that the tonoplast is absorbed ", unclear to me what the authors claim here.
  12. Lines 217-218: totally unclear statement. Please rephrase.
  13. Lines 240-242: unnecessary
  14. Line 248: "First, "
  15. Line 265: "in microautophagy, the autophagic bodies are directly surrounded by a membrane from the tonoplast." Furthermore, this sentence should be placed in the beginning.     

Author Response

Answers to Reviewer 2

[The indicated line numbers are those in the "final" view without markups.]

Major points:

(1) [Reviewer’s comment]

As aforementioned, the language is difficult to read due to poor English use. I suggest serious language improvement.

[Our response]

The English language of whole manuscript has been checked once again, although the proofreading had been done before the submission. We improved English in some parts of the article:

Lines 74-75 – the sentence has been corrected

Line 180, 185 – corrections of phrases

Line 219 – 222 - the sentence has been corrected

Line 223 – 229 - the sentence has been corrected

Line 231 – 237 the phrase has been moved from the bottom and modified (please also see the answer to the reviewer's last comment no.15).

Line 233  ‘by a membrane’

Line 252  ‘the capture of living samples’

Line 254 - ‘can suggest the origin of the macroautophagic vesicles’

(2) [Reviewer’s comment]

Authors do not present a critical evaluation of published results and in places text (described below in minor comments) is incomprehensible.

[Our response]

We corrected the text according to the suggestions listed in the minor points.

(3) [Reviewer’s comment]

In most cases, the description of the published data is swallowed.

[Our response]

We wanted to summarize the most important information about the data and results that comes from the past studies, because of the limitation of words dedicated to the review article. We revised the manuscript according to reviewers’ 2 suggestions (listed in the minor points) and the comments from reviewer 1 and the English editor.

(4) [Reviewer’s comment]

The text is very wordy and can be significantly reduced to improve readability.

[Our response]

We modified the main text together with English editor (as mentioned in our response to the major point 1). According to the minor pint from the reviewer 2 "Lines 242 - 245: the sentences have been deleted", the sentences from just before the Lines 238 have been deleted.

Also, the primary version of the manuscript had been shortened significantly before the submission. In our opinion the current version contains only the most important information and should not be reduced more.

(5) [Reviewer’s comment]

The message of which is the major and undisputed difference between macro- and microautophagy is not conveyed by the text. However, the image is well-prepared.

[Our response]

This review spotlights microautophagy, which is not fully focused on plants yet andmentions what is known or unknown in plant microautophagy. The basic difference between macro- and microautophagy is stated in the introduction, namely the information that the microautophagy membrane originates from the vacuolar membrane/tonoplast.

The manuscript also proposes several possible techniques to observe microautophagy in the section "3.2 Methods for...". As mentioned by reviewer 2, Line 265: "in microautophagy, the autophagic bodies are directly surrounded by a membrane from the tonoplast." is a key point to distinguish the autophagic bodies, and the origin of the autophagic body's membrane can be one of the clue to reveal that the vesicle is derived from macro- or microautophagy process. We agree to the last comment from reviewer 2 that this sentence should appear earlier. Because this point is strongly related to methodology, we moved this information to the beginning of the section "3.2 Methods for..." (Lines 231 - 237).

(6) [Reviewer’s comment]

The layout of the text could be better and the text better organized in distinct sessions with clearer messages (including critical evaluations in places-see also comment 2). 

[Our response]

We did try to modify the manuscript in the allottedperiod. Modifications we performed are as follows:

Lines 197, 199, 202: reference numbers were inserted.

Lines 219 – 222 and 223 - 227: the sentences have been rearranged for better understanding

Line 231: rearrangement of the phrase (please also see the answer to the reviewer's last comment no.15).

Line 238: As mentioned above, according to the minor pint from the reviewer 2 "Lines 242 - 245: the sentences have been deleted", the sentences from just before the Lines 238 have been deleted.

Minor points:

(1-4) [Reviewer’s comment]

Line 15: "In yeast and animals,"

Line 26: "developmental"

Line 52: "In this review, "

Line 63: "consequence of these existing studies, "

[Our response]

We corrected these points.

(5) [Reviewer’s comment]

Line 73: "Peroxisome degradation in the microautophagic manner is called micropexophagy", English here should be improved significantly.

[Our response]

We modified the sentence:

“The direct incorporation of peroxisomes into the vacuole by microautophagic invagination is defined as micropexophagy….” (Lines 74 - 75)

(6-8) [Reviewer’s comment]

Table 1: "Envelope", "Calmodulin", "shift-induced"

Line 177: "component involved in"

Line 182: "photodamaged chloroplasts are invaginated by the tonoplast and”

[Our response]

We corrected these points (Tabel 1, Line 180, 185).

(9) [Reviewer’s comment]

Line 203: Reference missing

[Our response]

We added a reference in Line 208 [71] (Goto-Yamada et al., 2019). Related this correction, references [73], [71] and [72] have changed to [71], [72] and [73], respectively (Lines 202, 208, 217, 222, 270, 272, 283 and the reference list Line 488-490).

(10) [Reviewer’s comment]

Line 209: "Arabidopsis roots and sucrose"

[Our response]

We modified this sentence.Line 215: ‘Arabidopsis roots and BY-2 cells in sucrose starvation’

(11) [Reviewer’s comment]

Line 213: "with E-64d, suggesting that the tonoplast is absorbed ", unclear to me what the authors claim here.

[Our response]

We corrected the sentence:

“Also, the amount of the tonoplast decreased in the cells treated with E-64d, suggesting that the tonoplast is taken up into the E-64d vesicles. This observation demonstrates that microautophagy process may contribute to the vacuolar formation.” (Lines 219 - 222)

(12) [Reviewer’s comment]

Lines 217-218: totally unclear statement. Please rephrase.

[Our response]

We agree that the original sentence was confusing. We wanted to highlight here that the contribution of the lipid raft on the vacuolar membrane is reported in microautophagy and lipid rafts are provided by macroautophagy(Müller et al., 2000; Tsuji et al., 2017), what was also explained in P3, Line 137-139.

We have modified the sentences to clarify this point:

“However, the contribution of these ATGs to microautophagy may be indirect because the accumulation of E64-d vesicles is not entirely stopped in atgmutants suggesting another, complementary pathway. As mentioned above, macroautophagy contributes to microautophagy by providing the lipid rafts required for the initiation of microautophagy on vacuolar membrane in yeast.” (Lines 223 – 227)

(13) [Reviewer’s comment]

Lines 240-242: unnecessary

[Our response]

We have deleted the sentences from just before the Lines 238..

(14) [Reviewer’s comment]

Line 248: "First, "

[Our response]

We corrected this point (Line 260).

(15) [Reviewer’s comment]

Line 265: "in microautophagy, the autophagic bodies are directly surrounded by a membrane from the tonoplast." Furthermore, this sentence should be placed in the beginning.

[Our response]

This part has been moved to the beginning of section "3.2 Methods for..." with some modifications(Lines 231 – 237).

Round 2

Reviewer 2 Report

Authors have improved their manuscript. One additional comment is listed below.

Line 56: Still, the authors need to refer to the model used to get the results. For example, in Line 100: unclear which organism the authors refer to. It is yeast.

Author Response

Response to Decision Letter

[The indicated line numbers are those in the final view without markups. The corrected parts during 2nd revision were highlighted in yellow.]

Our answers to the comments made by the Reviewer 2 are as follows:

Minor points

(1) [Reviewer’s comment]

Line 56: Still, the authors need to refer to the model used to get the results. For example, in 

Line 100: unclear which organism the authors refer to. It is yeast.

[Our response]

We added the information about model organisms used in the studies cited in our manuscript. 

Line 96: ‘in S. cerevisiae’

Lines 102 – 103: ‘in the yeast S. cerevisiae’

Line 245: ‘in barley’

Line 276: ‘in Arabidopsis’ 

(2) [Reviewer’s comment]

Moderate English changes required.

[Our response]

Adding commas in the following lines:

Line 31: ’… most well studied, and more than….’

Line 74: ‘…as micropexophagy, and a number of…’ 

Line 114: ‘…in yeasts, and most of…’

Line 228: ‘In macroautophagy, the outer…tonoplast, and then…’

Line 238: ‘… of Pinus densiflora, and it was observed…’ 

Line 258: ‘After incorporation, it is associated…’

Line 261: ‘…fluorescent dye, it can be…’

Line 282: ‘…autophagy, it is important…’. 

English language improvement:

Line 46: word replacement; ‘can be’ changed to ‘may consist of’.

Line 128: correction of ‘Drosophila melanogaster’.

Lines 132:  adding a dash in ‘freeze-fracture’.

Line 142: remove ‘, and there seems to be less generality’.

Table 1: adding ‘the’ before ‘vacuolar membrane’.

Line 179: word replacement; ‘involved’ changed to ‘included’.

Lines 181 - 184: The sentence "Nakamura et al. (2018) reported..." was divided into two sentences with a modification (Line 183). 

Line 276: adding ‘to be’.